# Ultrasonographic Imaging Protocol and Sonoanatomy of the Lumbar Spine in Healthy Dogs

**DOI:** 10.3390/ani12091187

**Published:** 2022-05-06

**Authors:** Justyna Abako, Piotr Holak, Joanna Bajon, Yauheni Zhalniarovich

**Affiliations:** Department of Surgery and Radiology with Clinic, Faculty of Veterinary Medicine, University of Warmia and Mazury, 10-719 Olsztyn, Poland; piotr-holak@wp.pl (P.H.); j_glodek@wp.pl (J.B.); eugeniusz.zolnierowicz@uwm.edu.pl (Y.Z.)

**Keywords:** anatomy, ultrasound (USG), canine, spinal cord

## Abstract

**Simple Summary:**

Ultrasonography is a popular imaging technique in veterinary medicine. It is also often used for imaging of the musculoskeletal system. In the current literature, there are descriptions of using ultrasonography to perform examination of the spine area in the dog, for example, to make perineural blocks or identification of the lesions in the intraoperative examination. Despite these applications, there is no protocol for imaging the spine of dogs, and there are few descriptions of its sonoanatomy. The aim of this paper is to present the protocol of examination of the lumbar spine in the dog and to describe sonoanatomy of this area.

**Abstract:**

Ultrasound is an imaging technique commonly used in veterinary medicine. Ultrasound devices are widely available, their means of examination are relatively short and cheap, and they do not generate ionizing radiation. In addition, ultrasound generally does not need to be performed under general anesthesia. This study was performed on 23 canine cadavers with full clinical histories and with no confirmed pathological changes in the spine region. The imaging modalities were established in dogs in lateral recumbency, with the selected side being the uppermost angle, in a neutral position. All dogs were examined in the transverse and longitudinal planes. Sacral crest, intertransverse ligament, vertebral canal floor, vertebral body, and intervertebral discs were only visible in the longitudinal plane. Vertebral arch, supraspinal ligament, dorsal wall of the vertebral canal and muscles were visualized only in the transverse plane. This article provides a brief and relatively easy-to-perform protocol for ultrasound imaging of the lumbar spine of dogs. In addition, it presents a detailed description of the sonoanatomy of the area under investigation.

## 1. Introduction

Diagnosis of the spine and spinal cord diseases is based mainly on clinical symptoms. However, diagnosis often requires advanced imaging techniques such as magnetic resonance imaging, computed tomography, and myelography. Unlike these imaging methods, access to ultrasound machines is much simpler. Furthermore, ultrasonography does not require general anesthesia and does not use ionizing radiation [1].

Intervertebral disc herniation is the most common disease of the thoracolumbar spine in dogs [2]. Proper determination of the intervertebral space in which the herniation occurred is a key point in planning the surgical procedure [3]. Visualization of the spine and spinal cord is difficult intraoperatively. During the surgical procedure, it is important to determine not only the location of the lesion but also its extent, in order to determine the appropriate access and thus proper decompression of the spine and subsequent prognosis [4]. The use of intraoperative imaging methods (such as fluoroscopy and portable X-ray) is also recommended in order to avoid erroneous determination of the intervertebral space, which potentially increases the need for the use of ultrasound in the assessment of the spine. As indicated by Nanai et al. (2007) and others, the intraoperative use of ultrasound allowed the diagnosis of diseases such as intervertebral disc herniation, cervical spondylomyelopathy, tumors of the spinal cord and nerve sheaths, and granulocytic myelitis [4,5,6,7,8]. In another study, the use of an ultrasound machine to determine the position of the intervertebral space allowed for the significant improvement in the accuracy and correctness of neurolocation compared to palpation [3].

In veterinary and human medicine, ultrasonography is commonly used to perform peripheral nerve blocks and it is becoming more and more popular as an adjunct method for anesthesia of the spinal nerves [9,10,11,12,13,14,15]. There are reports of ultrasound-guided cerebrospinal fluid collection, epidural catheter placement, and epidural puncture [16,17,18,19,20,21,22,23].

Ultrasonography shows great potential in the diagnosis of spinal diseases and as a supportive method in procedures performed on the spine. The aim of this article is to describe normal images of the lumbar spine as a comparative model in the diagnosis of spine diseases and to propose a clinically useful examination protocol.

## 2. Materials and Methods

This pilot study was performed from May 2020 to January 2021. Each examination was performed by one specialist (J.A.). Twenty-three canine cadavers with full clinical histories and with no pathological lesions in the spine, confirmed by X-ray examination, were used. The dogs were euthanized for reasons unrelated to this study. The mean age of the dogs was 11 years (5–17 years) and mean weight 12.88 kg (2–36 kg). There were 8 female (34.8%) and 15 male (65.2%) dogs. The breeds of the cadavers are listed in Table 1. The clinical history of the dogs was printed and also saved on compact discs and a portable hard drive.

The hair was clipped from each cadaver in the lumbosacral region, from the thoracic (T) 13 to sacral (S) 2 vertebrae, 1/3 of the length of the costal arch. Next, the skin in this area was covered with ethanol and ultrasound gel as an acoustic coupling agent. An ultrasound machine (Esaote MyLab Seven, Esaote, Italy) with a 13–3 MHz non-sterile linear probe was used for examination.

Examinations were performed in dogs in lateral recumbency, with the selected side being the uppermost angle in a neutral position. All dogs were examined in the transverse and longitudinal planes on both sides. Initially, the probe was applied perpendicular to the skin on the dorsal side in the longitudinal plane in the midline to identify the first sacral and the seventh lumbar vertebrae (Figure 1). The main landmarks are particular spinous processes of the lumbar vertebrae (counting from the 7th lumbar vertebra) and sacral crest. The probe was moved cranially and the consecutive vertebrae were identified by counting them from lumbar (L) 7 to T13. Next, the probe was rotated 90 degrees to the transverse plane to show interlaminar spaces (Figure 2). Then, it was rotated again to the longitudinal plane and slightly shifted to the paramedian line to show facet joints and vertebral laminae (Figure 3). The probe was also placed on the ventral side to visualize the vertebral bodies, intervertebral discs, and floor of the vertebral canal. (Figure 4). Images of identified anatomical structures and short films from the examinations were recorded and saved with the clinical data of each cadaver.

## 3. Results

The numbers of visible structures in each case are listed in Table 2 and Table 3. The sacral crest, intertransverse ligament, vertebral canal floor, vertebral body, and intervertebral discs were only visible in the longitudinal plane (Figure 5 and Figure 6). The vertebral arch, supraspinal ligament, dorsal wall of the vertebral canal and muscles (spinalis, semispinalis, multifidus, iliocostalis, longissimus) were visualized only in the transverse plane (Figure 7).

In the longitudinal plane, vertebral bodies and intervertebral discs were visible in all cadavers. The sacral crest and spinous processes were visible in only two cases. Paraspinal muscles were visible in all cases but could not be distinguished. In the transverse plane, paraspinal muscles were seen in all examinations. The spinalis and semispinalis muscles were distinguished in six cadavers, and the multifidus, iliocostalis, and longissimus were distinguished in seven cadavers. The vertebral arch and supraspinal ligament were visible in only one examination.

## 4. Discussion

This article provides a brief and relatively easy-to-perform protocol for ultrasound applied to the lumbar spine of dogs. In addition, it presents a detailed description of the sonoanatomy of the area under investigation. According to the authors’ knowledge, in the current literature there are few partial descriptions of the ultrasound anatomy of a dog’s spine [24,25,26,27,28], as the main purpose of these previous publications was to present the use of ultrasound as a technique to support and facilitate epidural puncture performance or paraspinal nerve blocks.

Based on the purpose of the studies described above, ultrasound was performed in dogs in the sternal recumbency, often with the pelvic limbs extended cranially [3,26,27]. Studies in which dogs were placed in lateral recumbency [24,28,29] did not involve an examination of the spine on the ventral side. According to the authors, examination in the lateral recumbency with limbs in a neutral position allows one to obtain an acoustic window in the longitudinal plane and show and evaluate anatomical structures such as vertebral bodies or intervertebral discs, which is difficult or impossible to perform in the examination in sternal recumbency. Graff et al. (2014) identified the body of the L6 vertebra and the intervertebral disc L6-7 on the abdominal side, but only to illustrate the spinal nerve roots running near them, without a detailed description of the sonoanatomy of the disc. MacKenzie et al. (2014) identified intervertebral discs in the spine segments most at risk of degenerative disc disease, but the purpose of this study was to select a site for injecting preparations into the nucleus pulposus of the disc.

As the mean age of the dogs studied was 11 years, degenerative changes in vertebrae and intervertebral discs could be expected. However, no such changes were observed.

According to human medicine studies, ultrasound identification has been shown to be more precise than palpation of landmarks for spinal neurosurgery. Identification of the correct intervertebral space is crucial in spinal surgery, and surgery performed in the wrong place is a rare mistake with serious consequences [3]. The use of ultrasound to locate intervertebral spaces prior to epidural injection has also been described in several publications [24,26,27]. The presented ultrasound protocol allowed us to determine the exact location of the individual lumbar vertebrae, intervertebral spaces, and sacrum in the medial longitudinal plane on the back side of the spine. According to Liotta et al. (2015), for the best visibility of the target area, a probe was located in the parasagittal plane. An additional reference point for determining the position of the probe was the wing of the ilium [27]. In this study, wing of the ilium was visible, but was not taken into account as a landmark or an element described as a paravertebral structure. In a study by Boursier et al. (2018), additional reference points were the spinal processes of the lumbar vertebrae and the last rib [3].

The intervertebral disc was a structure visible only in the longitudinal plane on the ventral side of the spine, as mentioned above. It was identified as a less echogenic, triangular, elongated area between the two hyperechogenic vertebral endplates. At times there was a hyperechoic line representing its border. The echogenicity of the intervertebral disc was variable. In some of the cadavers examined, intervertebral discs showed hyperechogenic lines parallel to each other and perpendicular to the long axis; in addition, the ventral border was more hyperechogenic. In addition, in some cases, intervertebral discs seemed to be more pointed above the vertebral bodies. According to Naish et al. (2003), the fibrous ring of the disc may consist of parallel, gently hyperechogenic lines; however, despite the linear structure being demonstrated, it cannot be reliably differentiated with the nucleus pulposus [30,31,32]. In this study, the echogenicity of the above-mentioned lines was variable. This may be related to the development of the degenerative changes in the intervertebral discs, which was not confirmed. It is important that the intervertebral disc provides an acoustic window for imaging the spinal canal. The floor of the spinal canal will then be visible as a hyperechogenic line of different thickness, parallel to the long axis of the spine.

Examination in the transverse plane is more difficult to perform than in the longitudinal plane due to the small acoustic window, but it also allows for easier identification of muscles. Muscles are iso- to hypoechoic structures, with heterogeneous echogenicity, surrounded by a linear hyperechoic fascia. They have been identified only in the transverse plane on the dorsal side of the spine. Lopes et al. (2018) described a similar ultrasound appearance of the muscles, but they also managed to describe the multifidus muscle in the longitudinal plane. In the authors’ current study, the intertransverse, interspinous, and rotators muscles were indistinguishable from the other muscle parts due to their small size.

The sonoanatomy and appearance of the other structures in the lumbar spine are similar to those described by other authors [24,27,28]. In the longitudinal sagittal plane on the dorsal side, the sacral crest was visible as a continuous hyperechoic line with an acoustic shadow, composed of three connected protrusions. Spinous processes of the lumbar spine were thin hyperechoic curved line with a posterior acoustic shadow. On the ventral side, the vertebral body was visible as a curved hyperechoic line with a long posterior acoustic shadow. The vertebral canal floor was observed as a thin hyperechoic line, located under the intervertebral discs.

In the longitudinal parasagittal plane, the articular processes lie above the transverse processes and are wider than them. The articular processes were visible as two hyperechoic lines, adjacent to each other (Figure 8). Transverse processes were narrow hyperechoic lines with a long posterior acoustic shadow. The intertransverse ligament was an isoechogenic area between the transverse processes, lying perpendicular to them and separated by two parallel, mildly hyperechoic lines (Figure 9).

In the transverse sagittal plane on the dorsal side, the vertebral arch was a thin hyperechoic line at the base of the spinous process and was difficult to distinguish. The supraspinal ligament was an oval-shaped structure and was also difficult to visualize due to its small size and homogenous echogenicity. Spinous processes were visible as long, narrow acoustic shadows. Articular processes were hyperechoic lines with a pointed acoustic shadow. Transverse processes were long hyperechoic lines with posterior acoustic shadow. The dorsal wall of the spinal canal was a thin, highly hyperechoic line, parallel to the course of the vertebral arch.

We were unable to identify the spinal nerves. Due to the nature of the study, vascular structures were also not visible. Their depiction is possible only in live dogs using Doppler techniques.

Moreover, in the larger and fatter dogs, the quality of the images obtained was definitely worse [16].

## 5. Conclusions

This publication presents a short and relatively easy-to-perform ultrasound protocol for the lumbar spine of dogs, allowing most anatomical structures to be identified and evaluated. As the examination was performed not only from the dorsal side of the spine but also from the ventral side, this allowed us to show the intervertebral discs and vertebral bodies. Due to the duration of the examination and the availability of equipment, ultrasound can be used as a technique to quickly assess the above-mentioned structures and to determine abnormalities at their sites. Further research in this area may help to determine the degree of mineralization and its effect on the onset of intervertebral disc disease. Since some of the images did not allow for the identification of all anatomical structures, the described imaging technique requires extensive training.

## Figures and Tables

**Figure 1 animals-12-01187-f001:**
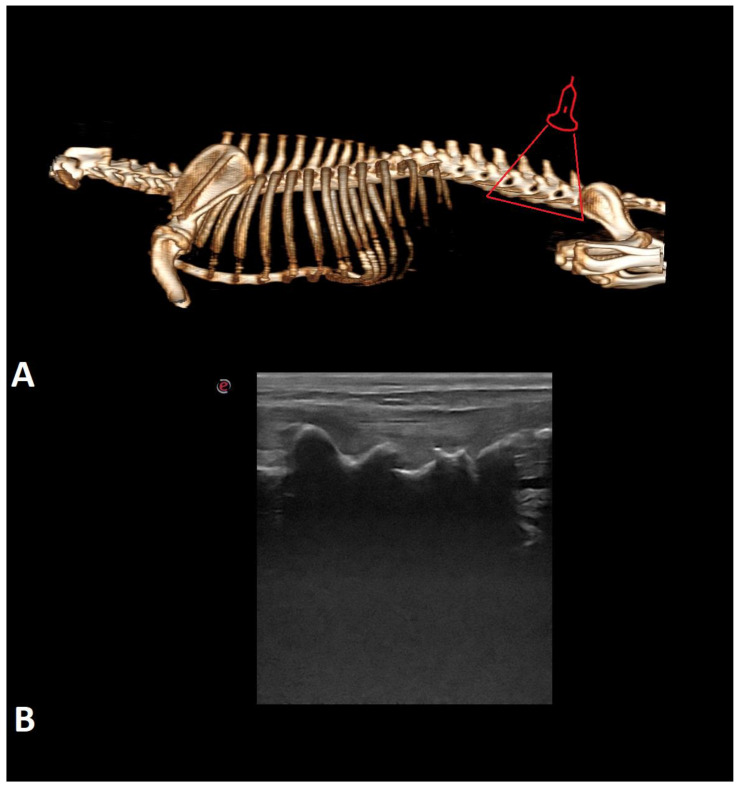
Three-dimension computed tomography (3D CT) scan (**A**) and ultrasonographic image along the longitudinal midline plane (**B**) of the lumbar spine of the dog. The red triangle indicates the location on the spine where the probe was applied.

**Figure 2 animals-12-01187-f002:**
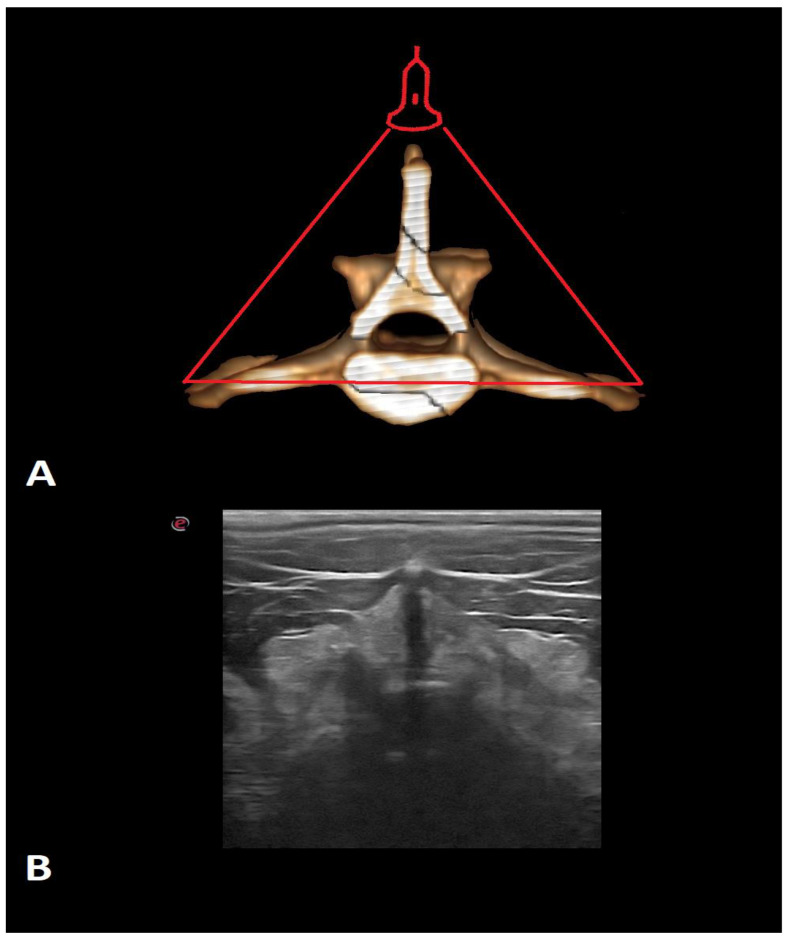
The 3D CT scan (**A**) and ultrasonographic image of the transverse midline plane (**B**) of the lumbar vertebra of the dog. The red triangle indicates the place on the spine where the probe was applied.

**Figure 3 animals-12-01187-f003:**
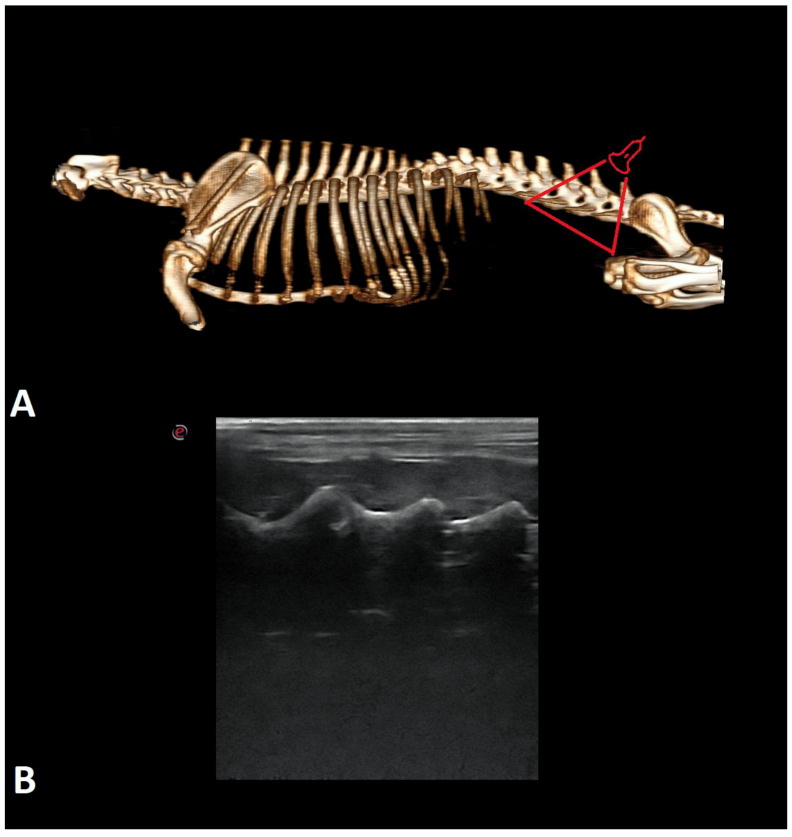
The 3D CT scan (**A**) and ultrasonographic image of the longitudinal paramedian plane on the dorsal side (**B**) of the lumbar spine of the dog. The red triangle indicates the place on the spine where the probe was applied.

**Figure 4 animals-12-01187-f004:**
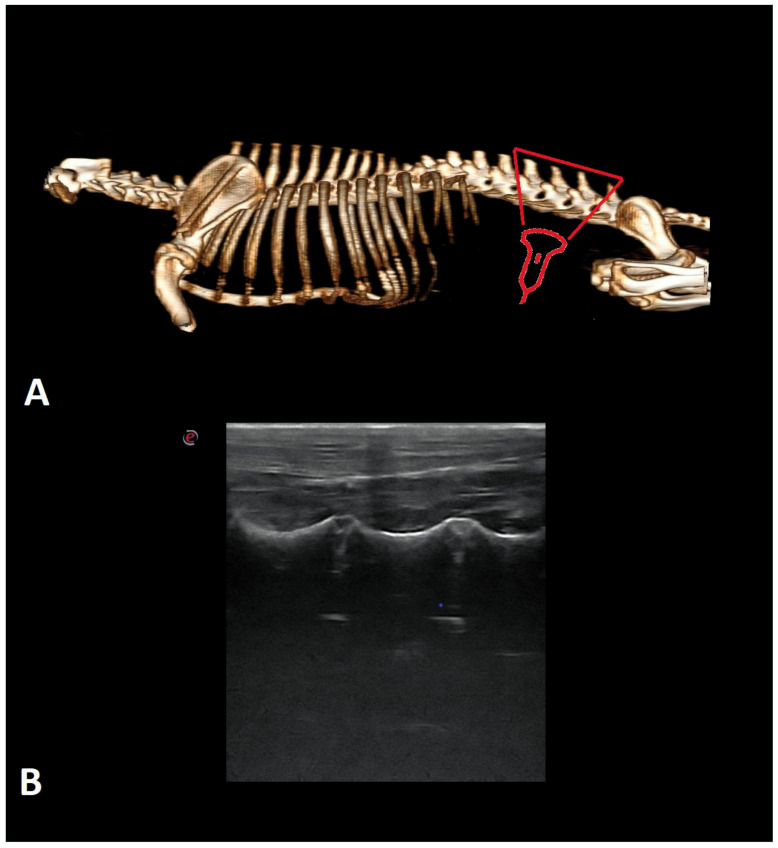
The 3D CT scan (**A**) and ultrasonographic image of the longitudinal paramedian plane on ventral side (**B**) of the lumbar spine of the dog. A red triangle indicates the place in the spine where the probe is applied.

**Figure 5 animals-12-01187-f005:**
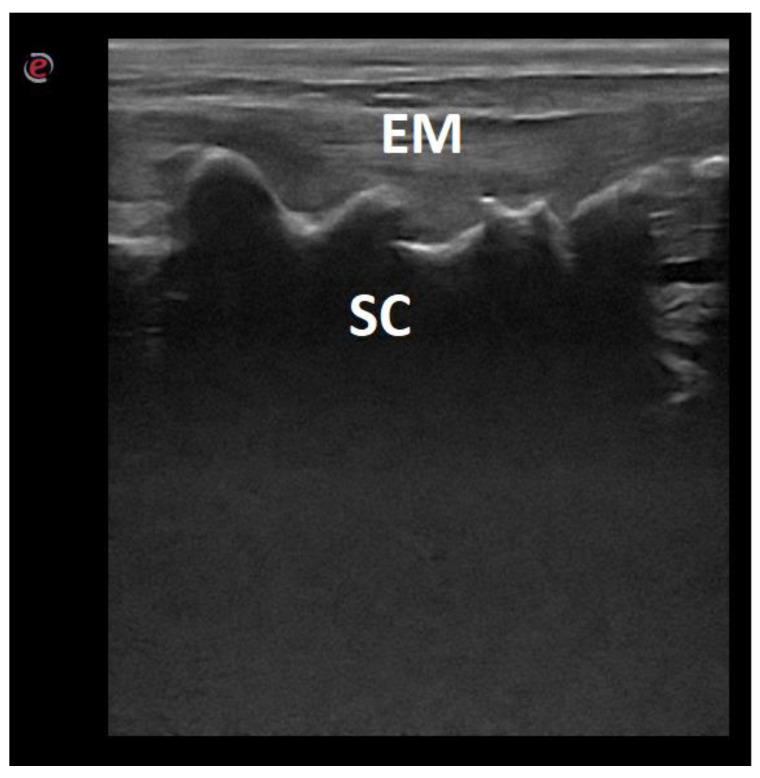
Ultrasonographic image of the lumbo-sacral spine of a dog in the longitudinal midline plane. EM—epaxial muscles; SC—sacral crest.

**Figure 6 animals-12-01187-f006:**
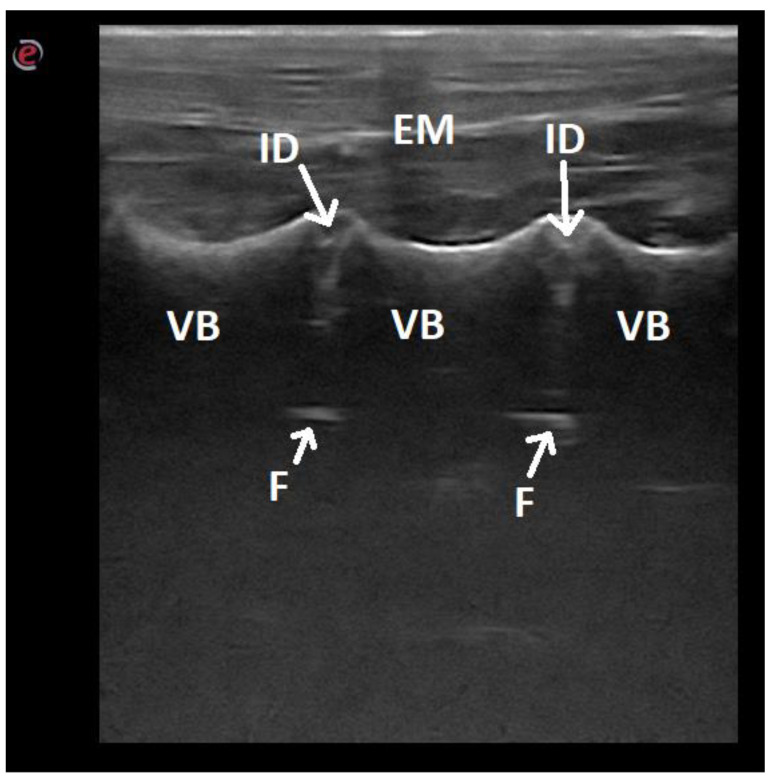
Ultrasonographic image of the lumbar spine of a dog in the longitudinal paramedian plane on the ventral side: EM—epaxial muscles; ID—intervertebral disc; VB—vertebra body; F—floor of the vertebral canal.

**Figure 7 animals-12-01187-f007:**
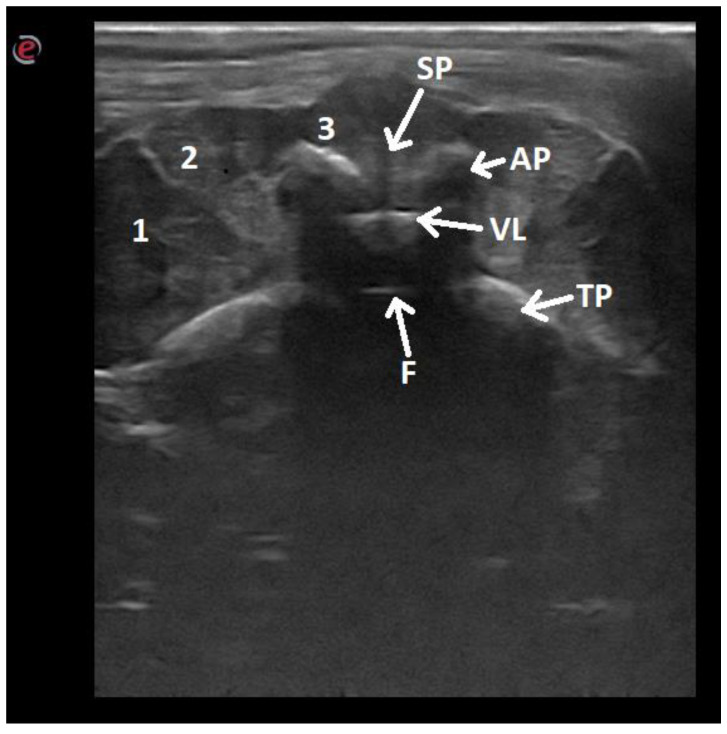
Ultrasonographic image of the lumbar vertebra of a dog in the transverse midline plane: 1—iliocostalis and longissimus muscles; 2—spinalis and semispinalis muscle; 3—multifidus muscle; SP—spinal process; AP—articular process; VL—vertebral lamina; TP—transverse process; F—floor of the vertebral canal.

**Figure 8 animals-12-01187-f008:**
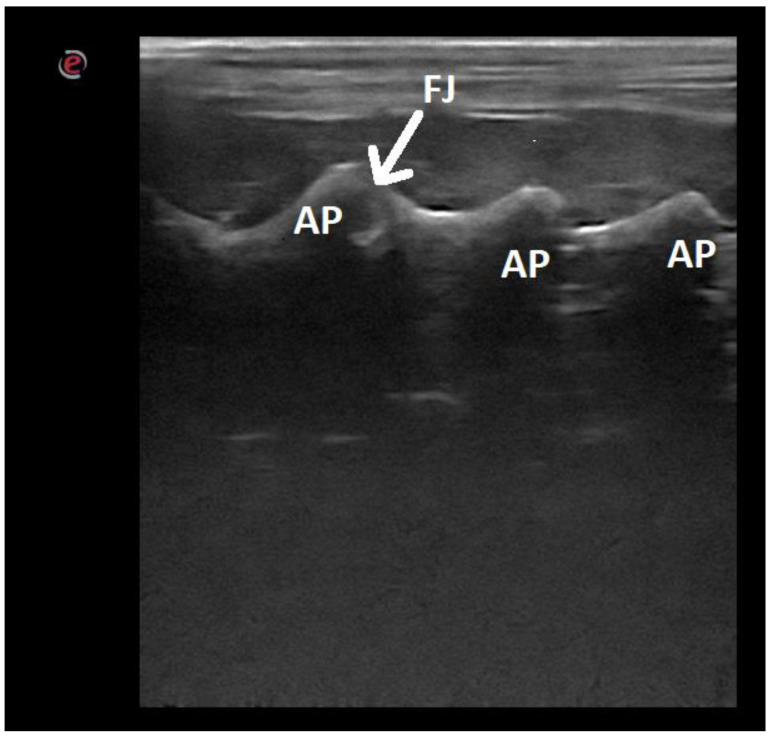
Ultrasonographic image of the lumbar spine of a dog in the longitudinal paramedian plane on the dorsal side: AP—articular process; FJ—facet joint.

**Figure 9 animals-12-01187-f009:**
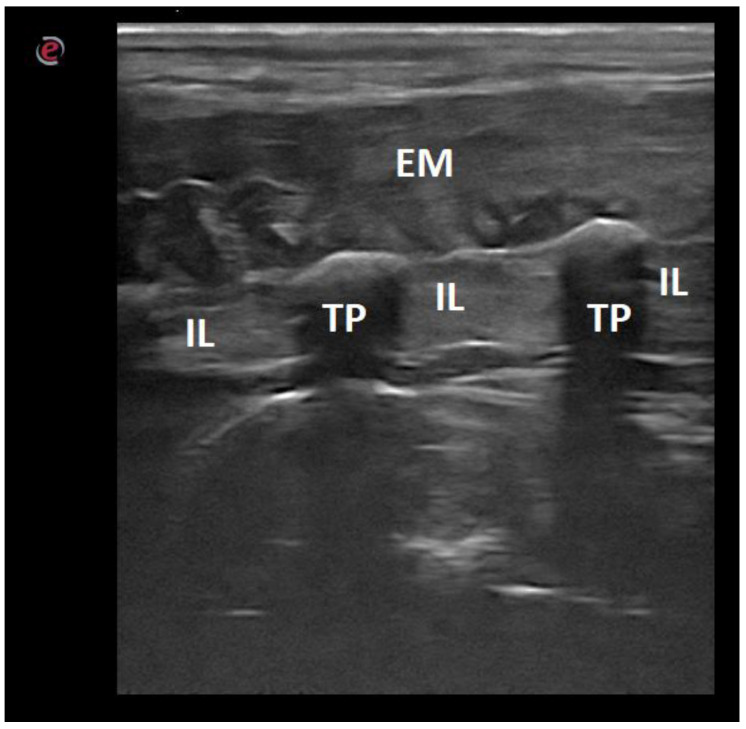
Ultrasonographic image of the lumbar spine of a dog in the longitudinal paramedian plane on the lateral side: EM—epaxial muscles; IL—intertransverse ligament; TP—transverse process.

**Table 1 animals-12-01187-t001:** The breeds of the cadavers used in the study.

Breed	Number of Cases
Mixed Breed	9
Yorkshire Terrier	3
German Shepherd	2
Labrador Retriever	2
Chihuahua	1
Cocker Spaniel	1
Dachshund	1
Irish Seter	1
Maltese	1
Miniature Schnauzer	1
Polish Hunting Dog	1

**Table 2 animals-12-01187-t002:** Structures identified in the study in the longitudinal plane and the number and percentage of their visibility in the examined cadavers.

Structure	Visibility (Number/% of Cases)
Sacral crest	2/8.7
Spinous processes	2/8.7
Intertransverse ligament	10/43.48
Vertebral canal floor	11/47.83
Articular processes	12/52.17
Transverse processes	15/65.22
Vertebral body	23/100
Intervertebral discs	23/100

**Table 3 animals-12-01187-t003:** Structures identified in the study in the transverse plane and the number and percentage of their visibility in the examined cadavers.

Structure	Visibility (Number/% of Cases)
Vertebral arch	1/4.35
Supraspinous ligament	1/4.35
Dorsal limit of the vertebral canal	4/17.39
Transverse processes	6/26.09
Spinalis and semispinalis muscle	6/26.09
Articular processes	7/30.43
Multifidus muscle	7/30.43
Iliocostalis and longissimus muscles	7/30.43
Spinous processes	9/39.13

## Data Availability

Data and materials are available from the corresponding author on reasonable request.

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
