# Peer review of "Ultrasonographic Imaging Protocol and Sonoanatomy of the Lumbar Spine in Healthy Dogs"

_animals, 2022, doi:10.3390/ani12091187_

Round 1

Reviewer 1 Report

Abstract

Line 20

How did you confirmed that no pathological changes were present in the spine region? Please clarify

Line 21

Did you performed the scan on both lateral recumbency? Did you scan both sides to see if any ease in the approach or difference in image quality or visualised structure?

Material and methods

  • How can you rule out bony changes? Did you take radiographs of the lumbar spine? Ultrasound is not a good imaging modality to determine small amount of new bone formation
  • Moreover, radiographs should have been used to rule out any anomalies in lumbar spine: congenital abnormalities, outnumber (8 or 6 lumbar vertebrae)
  • Ultrasound is operator dependent. Please clarify if the same person performed the scan in all 23 dogs. 
  • Images would benefit from labelling, arrow etc? Please add in the ultrasound image which vertebral body are you looking at
  • Can you estimate the inclination angle of the probes for the paramedian plane? Any landmark the authors considered while scanning?

Results

Line 115 and 116

Did the authors noted any difference in image quality, details and structure visibility between patient size and weight? Any correlation?

Subcutaneous fat might compromise the image quality and visibility of structure in ultrasound

Discussion

Line 175

How can you rule out bony changes? Did you take radiographs of the lumbar spine? See comments above

Line 183

Please clarify what you mean for “back side” of the spine

Line 242

This is an unclear statement. The spinal cord was not described and/or visualise, therefore it is incorrect to say that advanced imaging techniques are not necessary the detect lesions in the spine. Please clarify

Reviewer 2 Report

Is a well-structured article, understandable and legible. However, the literature used is minimal. I would like to make a few remarks.

L 51-55.  I consider that perineural blocks is another technique performed in a different anatomical location, it is not related to the specific issue and there is no reason to mention it.

L 90-94, figure 1. As is well known, in an ultrasound image, the tissue of interest should be in the center of the B-mode image. Ιn the caption it is mentioned «ultrasonographic image along the longitudinal midline plane (B) of the lumbar spine of the dog». the image must be changed, the lumbar spine in this image is in the lower third, almost not visible

Figure 2, image B : the B-mode image has many artifacts, please replace it

Figure 1-4: the linear transducer produce a more square or rectangular image. The image you indicate in red is one that can produce a curvilinear (microconvex) probe. Please change the red lines.

Additionally, in Materials and Methods is not mentioned the settings used. It is necessary  to  make an extensive report of the  scanning depth, specific frequency, gray map, focuses, gain, as the article describes an ultrasonographic imaging protocol.
